# Quadruple C-H activation coupled to hydrofunctionalization and C-H silylation/borylation enabled by weakly coordinated palladium catalyst

Bo-Cheng Tang[1], Wen-Xuan Lin[2], Xiang-Long Chen[1], Cai He[1], Jin-Tian Ma[1], Yan-Dong Wu[1], Yu Lan [2,3✉] & An-Xin Wu [1✉]

Unlike the well-reported 1,2-difunctionalization of alkenes that is directed by classic pyridine and imine-containing directing groups, oxo-palladacycle intermediates featuring weak Pd-O coordination have been less demonstrated in C-H activated cascade transformations. Here we report a quadruple C-H activation cascade as well as hydro-functionalization, C-H silylation/borylation sequence based on weakly coordinated palladium catalyst. The hydroxyl group modulates the intrinsic direction of the Heck reaction, and then acts as an interrupter that biases the reaction away from the classic β-H elimination and toward C-H functionalization. Mechanistically, density functional theory calculation provides important insights into the key six-membered oxo-palladacycle intermediates, and indicates that the β-H elimination is unfavorable both thermodynamically and kinetically. In this article, we focus on the versatility of this approach, which is a strategic expansion of the Heck reaction.

[1] Key Laboratory of Pesticide & Chemical Biology, Ministry of Education, College of Chemistry, Central China Normal University, Wuhan 430079, PR China. [2] College of Chemistry and Institute of Green Catalysis, Zhengzhou University, 100 Science Avenue, Zhengzhou 450001 Henan, PR China. [3] School of Chemistry and Chemical Engineering, and Chongqing Key Laboratory of Theoretical and Computational Chemistry, Chongqing University, Chongqing 400030, PR China. ✉email: lanyu@cqu.edu.cn; chwuax@mail.ccnu.edu.cn

The Heck reaction has become one of the most versatile synthetic tools because of its remarkable tolerance to a variety of functional groups and substrates[1–5]. In the past few years, studies about the classic Heck reactions gradually faded out of the vision of researchers while the derived Heck reaction take the lead[6–11]. Among the latter, further transformations of Heck-type migratory insertion intermediates have attracted much attention because combining the Heck process with other transformations provides a powerful synthetic strategy toward diverse and complex compounds. To reach this goal, subsequent β–H elimination should be interrupted; it is a typical side reaction in many cascade and 1,2-difunctionalization transformations, and is extremely hard to avoid because of favorable kinetics[12]. A typical solution for avoiding β–H elimination is to introduce a blocking group at the β-carbon position[13–16]. Other alternatives include adding a hydrogen source to afford a reductive Heck product[17–24], introducing norbornene derivatives in the Catellani reaction[25–28], producing a π-allyl/benzyl intermediate[29–31], and oxidizing the Pd (II) to Pd(IV) to create practicable methods[32–37]. Significantly, bidentate nitrogen directing groups have been well established that prevent β–H elimination in 1,2-difunctionalization and hydrofunctionalization reactions via a thermodynamically stable metallacycle[38–46]. However, thermodynamically stable intermediates engendered by strongly chelating functional groups diminish their reactivity in the subsequent functionalization step[47,48]. This decreased reactivity limits the possibilities for cascade transformations, and thus the reactions are limited to mono- and difunctionalization (Fig. 1a). Therefore, we sought a functional group that has relatively weak coordinating properties that is also present in common substrates.

Phenol derivatives are common and typical substrate-directable compounds[49–52]. However, their use as directing groups has only been developed recently[53–57], probably because of their weaker σ-donating properties compared with classic imine and pyridine-containing directing groups[47,58–60]. Apparently, this weaker σ-donation increases the reactivity of the metallacycle intermediate and the difficulty of the first functionalization step. Therefore, we envisioned a cascade reaction initiated by a Heck-type process, rather than coordination, to avoid low-efficiency functionalization. Subsequently, the judiciously positioned hydroxyl group stabilized the σ-alkylpalladium intermediate, which is less thermodynamically stable and highly reactive compared to strongly chelating metallacycle complexes, and thereby suppressed β–H elimination and enabled the multi-component cascade reaction (Fig. 1b). Herein, we report a weaker coordination dominated Pd (0)-catalyzed quadruple C–H activation cascade as well as hydrofunctionalization, C–H silylation, and C–H borylation sequences. By design, the hydroxyl group acts as a controller that dominates the reaction after modulating the intrinsic direction of the Heck reaction.

## Results

**Study on quadruple C–H activation.** After optimizing the reaction conditions (Supplementary Tables 1 and 2), we examined the substrate scope of quadruple C–H activation reaction. We first investigated the scope of the alkenes (Fig. 2a). Various 2′-OH-acetophenone substrates ($R^1$) bearing an electron-donating group, a halogen group, or a multiply substituted group all reacted smoothly to provide the corresponding products in high yields (**5–19**). Next, we investigated the effect of the substituent $R^2$. As predicted, various alkyl, halogen, electron-withdrawing, and electron-donating groups were also tolerated. Note that ortho-, meta-, and multiply substituted $R^2$ were all smoothly converted into final products in satisfactory yields (**20–46**). Nevertheless, fused-ring substituents, such as naphthalen-2-yl and phenanthrene-9-yl, were not compatible with our transformation, and afforded an inseparable complex mixture under the standard reaction conditions. Conversely, heterocyclic substituents showed excellent reactivity, furnishing the desired products in good yields (**47–48**).

A series of aryl iodide derivatives were subsequently evaluated with this protocol (Fig. 2b). Contrary to our expectations, the reaction was somewhat sensitive to reactants with larger

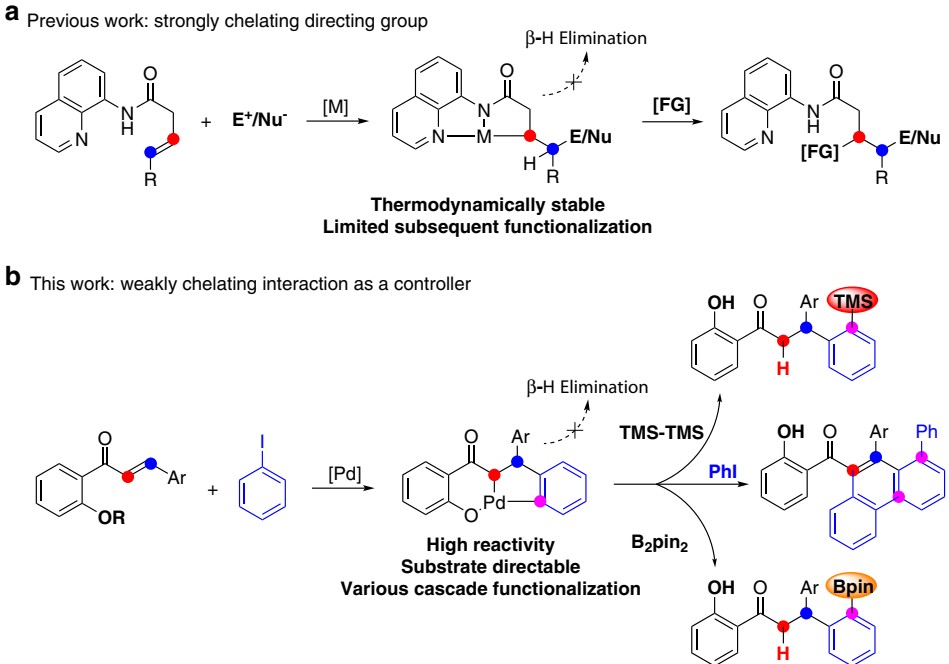

**Fig. 1 Transformation of Heck-type intermediate controlled by directing functional groups. a** Transition-metal catalyzed di-functionalization of alkene enabled by strongly chelating directing group. **b** Multi-functionalization enabled by weakly chelating interaction.

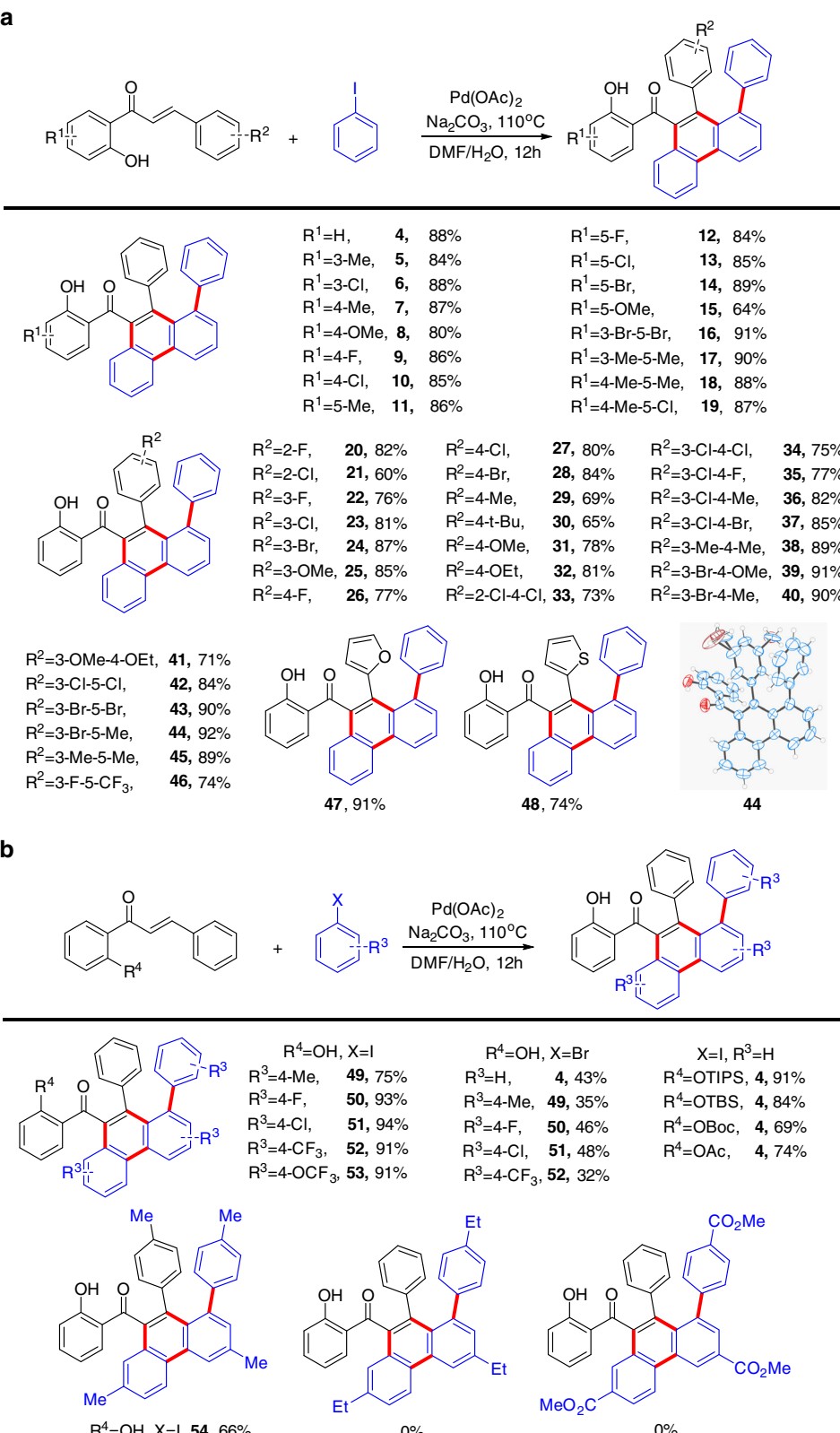

**Fig. 2 Substrate scope in quadruple C–H activation cascade. a** Substrate scope of alkene. **b** Substrate scope of aryl halide and protecting group. Reaction conditions: alkene (0.1 mmol), aryl halides (0.4 mmol), Pd(OAc)$_2$ (5 mol%), Na$_2$CO$_3$ (0.2 mmol), DMF (2 ml), H$_2$O (40 μl), 110 °C, 12 h. Yields of isolated products based on alkene.

substituents, such as -4-Et and -4-CO₂Me; reactivity was poor and resulted in a complex mixture or no reaction. Gratifyingly, the reaction conditions were sufficiently mild to deliver high yields even with electron-withdrawing and halogen groups, which provides possibilities for further functionalization (**49–53**). Aryl bromides also afford the corresponding products, although in much lower yield. Substrates containing functional groups derived from the hydroxyl group of **1** (R⁴ = OTIPS, OTBS, OBoc, OAc) were also investigated. Quite interestingly, these derivative substrates afforded products that were deprotected in situ, in one instance (R⁴ = OTIPS) in an even higher yield than the model reaction. The molecular structure of products **4**, **44**, and **50** were verified by single-crystal X-ray diffraction analyses.

**Study on C–H silylation**. The putative six-membered oxo-palladacycle intermediate would be challenging to isolate due to its thermodynamic instability[47]. However, we envisaged that this highly reactive species could be captured by other synthetically versatile reagents under our strategy. Encouraged by the extensive investigations of silylation and borylation in transition-metal-catalyzed C–H activation as well as the subsequent powerful transformations of organosilicon/organoboron compounds[61–64], we decided to apply our strategy to synthesize a series of silicon- and boron-containing compounds. We first investigated C–H silylation in our cascade reaction with hexamethyldisilane as the model substrate. Gratifyingly, compound **69** was isolated in 78% yield. Subsequent screening of the reaction conditions with the addition of 1eq of Me₄NOAc gave a remarkable increase of yield (Supplementary Table 3), a result similar to that reported by Zhang[65]. Notably, this high-efficiency transformation was nearly quantitative at 70 °C. Besides, control experiments revealed that the added hydrogen mainly came from water (Supplementary Figs. 7–9).

We then set out to determine the substrate scope of the C–H silylation protocol. Substitutions of the alkene moiety were first examined (Fig. 3a). Gratifyingly, common substituents including alkyl, phenyl, halogens, electron-donating groups (**55–71**), as well as thienyl and ferrocene groups (**72–73**) were well-tolerated. The molecular structure of product **73** was verified by X-ray single-crystal diffraction analysis. Next, we extended this silylation from alkenes to aryl iodide substrates (Fig. 3b). Surprisingly, this cascade process exhibits an extremely efficient transformation and excellent substrate compatibility for various aryl iodide moieties, with nearly quantitative yields for many of the substrates (**74–100**). Remarkably, compounds **101** and **102** could be obtained directly from the corresponding aryl iodide derivatives, although in relatively low yield.

**Study on C–H borylation**. Organoboron building blocks are highly versatile, particularly in the organic materials and pharmaceutical fields, with researchers making dramatic progress on C–H borylation reactions in the past few decades[66–68]. Considering the success with Rh and Ir-catalyzed C–H borylation, and comparatively fewer reports of Pd(0)-catalyzed cascade reactions[69–72], we subsequently investigated the possibility of C–H borylation under our strategy. After screening a series of reaction conditions, we found that bis(pinacolato)diboron was an effective boron source. This transformation afforded a satisfactory yield under similar reaction conditions, albeit at a higher temperature, compared with silylation (Supplementary Table 4).

Next, we probed the substrate scope of this C–H borylation process (Fig. 4), various aryl iodides were evaluated first. High yields of the corresponding products were obtained for *ortho*- and *para*-substituted aryl iodide derivatives (**104–119**), especially for substrates containing electron-donating-groups (**109, 118, 119**).

Conversely, substituted aryl iodides containing strongly electron-withdrawing groups (such as trifluoromethyl-, nitro-, and polyhalogen-) were not compatible, which resulted in inseparable complex mixtures or no reaction. We then investigated several representative alkenes using 2-iodotoluene as a representative aryl iodide. Although the average yields of these compounds were comparatively lower (**120–128**), the heterocyclic aldehyde substrates exhibited a satisfactory result (**129–131**). The exact structures of products **103, 105**, and **131** were verified by X-ray single-crystal diffraction analysis.

Overall, these transformations exhibit extremely high selectivity and excellent compatibility with a wide variety of functional groups. No simple β–H elimination products were detected for any of the examples in Figs. 2–4, and it thus represents a powerful strategic expansion of the Heck reaction.

**Further study**. The potential synthetic applications of these cascade reactions were subsequently investigated, starting with a gram-scale experiment for the quadruple C–H activation cascade. The yield was slightly decreased at the 5- and 10-mmol scale under the standard conditions, affording product **4** in 72 and 60% yields, respectively. The hydroxyl group was easily further transformed into other functional groups. Compounds **136** and **137** were efficiently obtained in 90 and 85% yields, respectively (Fig. 5a, b). Interestingly, the fluoro-substituted product presented excellent reactivity with the carbazole derivatives and gave an almost quantitative yield of **138, 139**, and **140** (Fig. 5c). Moreover, the emission spectra of **138** and **139** presented a notable solvatochromic effect. A distinct red shift was observed, with increasing solvent polarity (430 nm in n-hexane to 520 nm in DMSO for **138**; 430 nm in n-hexane to 505 nm in DMSO for **139**), which is a typical CT (charge transfer) characteristic in the excited state (Fig. 6)[73,74]. Further studies toward CT and possible ESIPT (excited state intramolecular proton transfer) properties of these compounds are currently underway.

**Mechanistic investigation**. To gain insight into the mechanism of this domino reaction, a series of control experiments were conducted (Fig. 7). First, we investigated whether the hydroxyl group was necessary under standard conditions. No reaction occurred when we replaced the −OH with −H, −OMe, −OBu, −OBn, or −OMEM, which indicated that the hydroxyl group is necessary. Similarly, an inseparable complex mixture was obtained when the -OH was replaced with −NH₂ or −NHAc, which suggests that this transformation is highly specific to the hydroxyl group. In addition, we investigated the influence position of the HO-group on the aromatic ring on the reaction. As expected, only the *ortho*-substituted substrate afforded a smooth reaction whereas the *meta*- and *para*-substituted substrates gave Heck products in addition to unreacted starting materials, which strongly suggests an intramolecular coordination between the hydroxyl group and the metal center.

As previously discussed, replacing –OH with groups such as −OTIPS, −OTBS, and −OBoc gave deprotected final products, indicative of a probable six-membered oxo-palladacycle intermediate and an X-type coordination mode (Fig. 7a)[55,75]. To further investigate the potential coordination modes of the hydroxyl group with the palladium center, the cesium salt of **1** was investigated as substrate under the standard reaction conditions. Compound **4** was isolated in 86% yield (Fig. 7b), which suggested that the hydroxyl group serves as an anionic ligand. Interestingly, when we added 5 equiv. of n-BuCl to the reaction, we did not detect product, and instead, compound **141** was isolated in 13% yield along with 62% nucleophilic attack product. The exact structure of **141** was verified by X-ray single-crystal diffraction analysis. This control experiment

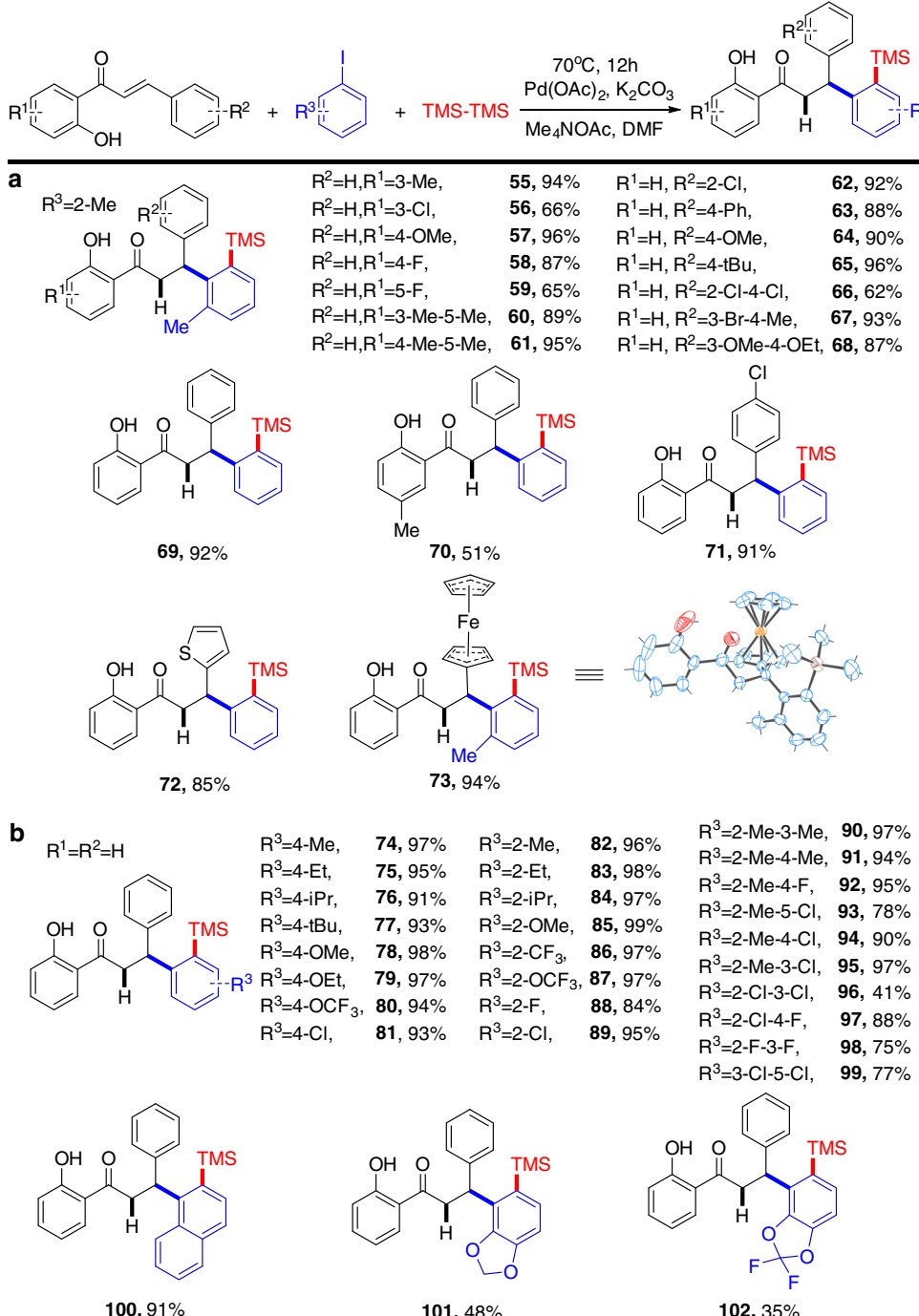

**Fig. 3 Substrate scope in cascade silylation. a** Substrate scope of alkene. **b** Substrate scope of aryl iodide. Reaction conditions: alkene (0.1 mmol), aryl iodide (0.15 mmol), hexamethyldisilane (0.2 mmol), Pd(OAc)$_2$ (5 mol%), K$_2$CO$_3$ (0.15 mmol), Me$_4$NOAc (0.1 mmol), DMF (2 ml), 70 °C, 12 h. Yields of isolated products based on alkene.

revealed that dehydrogenation is probably the last step of the entire transformation. Compound **141** probably resulted from a long-lived intermediate that is captured by n-BuCl before dehydrogenation (Fig. 7c). Submission of **141** to the standard reaction conditions delivered a complex mixture and significant amounts of recovered starting material. This suggested that the hydroxyl group is important to the dehydrogenation step (Fig. 7d).

To verify our hypothesis, deprotected compound **142** was prepared. Gratifyingly, transformation of **142** under standard reaction conditions (3 h) gave **4** in almost quantitative yield.

However, if the base was left out under otherwise standard conditions, only 28% yield of **4** (12 h) was obtained. The yield of **4** also clearly decreased upon removal of PhI from the standard reaction conditions. Only trace amounts of **4** could be detected upon removing the palladium catalyst, irrespective of whether an Ar or O$_2$ atmosphere was used (Fig. 7e). In summary, the hydroxyl group, palladium catalyst, and base are essential to the dehydrogenation process. The excess PhI may act as a potential oxidant of the palladium catalyst, as per GC-MS analysis. The lack of hydrogen gas evolution further supports this mechanism

**Fig. 4 Substrate scope in cascade borylation.** Reaction conditions: alkene (0.1 mmol), aryl iodide (0.15 mmol), bis(pinacolato)diboron (0.2 mmol), Pd(OAc)$_2$ (5 mol%), PivOK (0.2 mmol), DMF (2 ml), 100 °C, 12 h. Yields of isolated products based on alkene.

(Supplementary Figs. 1 and 2). Next, when **143** was used as substrate, no desired product was obtained (Fig. 7f), which indicates that the electronics of double bond, as well as the potential intramolecular hydrogen bonding interactions between the hydroxyl and carbonyl groups may be important in the reaction. In addition, the possibility of coordination between palladium center and carbonyl group should be taken into consideration. Finally, a series of isotopic labeling experiments were examined and the outcomes indicated that the hydrogen atom of the reconstituted hydroxyl group of the final product is derived from various moieties (Supplementary Figs. 3–6). An intramolecular hydrogen atom transfer process was not involved in this transformation (for selected control experiments of C–H silylation, see Supplementary Fig. 10).

**Theoretical calculation study**. Based on the previous studies about the Saegusa-Ito oxidation[76–79] and our experimental findings, we performed density functional theory calculations to further complement the mechanism of the final dehydrogenation (Fig. 8). According to the calculations, the Pd$^{II}$ species that is oxidized by excess PhI coordinates with **142** to form complex **IV**, followed by ligand metathesis to give complex **V** via a four-membered cyclic transition state with a 25.3 kcal mol$^{-1}$ free energy barrier (**IV-V**). In light of the essential nature of the

carbonyl group during the reaction (Fig. 7f), we proposed that the possibility of coordination between palladium center and carbonyl group should not be overlooked in the process of dehydrogenation. Under this circumstance, after a ligand exchange from complex **V**, two different hydrogen abstraction pathways are taken into consideration. First, the palladium center may coordinate with two acetates that act as monodentate ligand, then α-H abstraction may occur by the proximal acetate via **TS′$_{VI-VII}$** (Path B), along with the extrusion of one HOAc, followed by isomerization to afford complex **VII**. The activation energy of this pathway is calculated to be 32.9 kcal mol$^{-1}$ with respect to **VI** (**VI-VII**). Second, the possibility of hydrogen abstraction by free acetate ion could not be excluded. In this case, α-H abstraction of complex **VI** may occur via **TS$_{VI-VII}$** by the free acetate ion (Path A), followed by isomerization to form complex **VII**. Comparatively, this pathway requires an activation energy of 20.5 kcal mol$^{-1}$ with respect to **VI** (**VI-VII**), and 12.4 kcal mol$^{-1}$ lower than that of the former pathway. Upon the tautomerism of **VII** from enol to keto-form, a more stable complex **VIII** (15.8 kcal mol$^{-1}$ more stable than **VII**) is formed by ligand exchange. Then a *syn*-β-H elimination affords a Pd(II)-hydride complex **IX**. Interestingly, the direct O−H reductive elimination need to bear a free energy barrier of 28.3 kcal mol$^{-1}$. Alternatively, anionic complex **IX** can form a

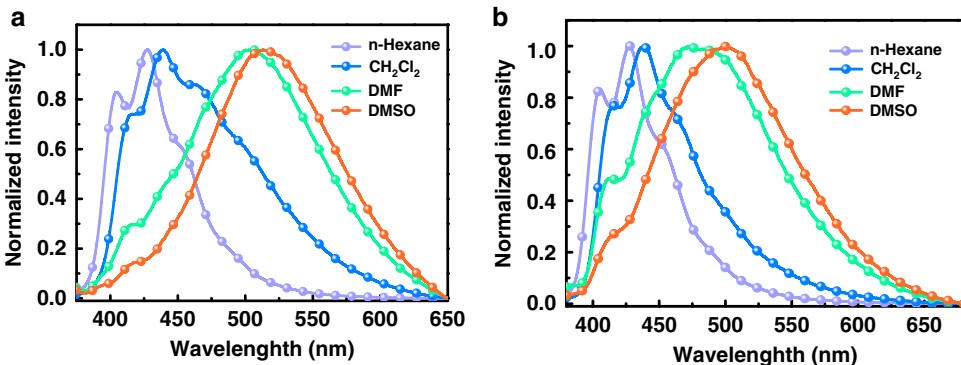

**Fig. 5 Further transformations. a** 1) Et$_3$N (2 eq), Tf$_2$O (2 eq), DCM, r.t. 2) Pd(PPh$_3$)$_4$ (5 mol%), LiCl (2 eq), Bu$_3$SnC$_2$H$_3$ (1.2 eq), DMF, Ar, 70 °C, 12 h. **b** PhI (1.5 eq), CuI (10 mol%), N,N-dimethylglycine (22 mol%), K$_3$PO$_4$ (2 eq), DMSO, Ar, 100 °C, 12 h. **c** Carbazole derivatives (3 eq), t-BuOK (3 eq), DMF, Ar, 150 °C, 72 h. All yield are isolated products.

**Fig. 6 Emission spectra. a** Normalized emission spectra of **138** in different solvents. **b** Normalized emission spectra of **139** in different solvents.

hydrogen bond with acetic acid to form complex **X** reversibly. Sequentially, a proton transfer takes place via transition state **TS$_{X-II}$**. The calculated free energy barrier for this step is only 15.2 kcal mol$^{-1}$. After ligand exchange, product **4** could be yielded with the regeneration of catalytically active complex **II**.

To further understand the formation of the key six-membered oxo-palladacycle intermediate in catalytic cycle, a corresponding computational study was conducted (Fig. 9). The formation of complex **V′** involves oxidative addition and *syn*-migratory insertion steps. The higher thermodynamic stability of complex **V′** may be partially attributed to the coordinated interaction between hydroxyl and palladium center. The process from **1** to **V′** is both feasible (ΔG$^{\neq}_{max}$ = 18.6 kcal mol$^{-1}$) and favorable (ΔG = −1.7 kcal mol$^{-1}$). In terms of substrate scope, no simple β–H

elimination products were detected for any of the examples in Figs. 2–4, presenting an efficient interruption of the classic Heck reaction pathway. Therefore, we hoped to elucidate the origin of this selectivity by calculating both pathways. According to the calculations, the β–H elimination occurs via **TS$_{V′-VII′b}$** and requires an activation energy of 9.8 kcal mol$^{-1}$ with respect to complex **V′**, and the corresponding product **VIII′b** is formed with an endergonicity of 3.3 kcal mol$^{-1}$ with respect to complex **V′**. Comparatively, the energy barrier for the formation of six-membered oxo-palladacycle via **TS$_{VI′-VII′a}$** is only 6.6 kcal mol$^{-1}$ with respect to complex **V′**, 3.2 kcal mol$^{-1}$ lower than that of the competing β–H elimination pathway. Besides, the corresponding complex **VII′a** is formed with an exergonicity of 14.3 kcal mol$^{-1}$ with respect to complex **V′**. More importantly, complex **VII′a** is

**a**

| R | Yields of product (%) | R | Yields of product (%) |
|---|---|---|---|
| H | N.R. | 2-OMEM | N.R. |
| 2-NH$_2$ | N.D. | 2-OTIPS | **4**, 91% |
| 2-NHAc | N.D. | 2-OTBS | **4**, 84% |
| 2-OMe | N.R. | 2-OBoc | **4**, 69% |
| 2-OBu | N.R. | 2-OAc | **4**, 74% |
| 2-OBn | N.R. | 3-OH | N.D. |
| 4-OH | N.D. | | |

**b**

**c**

**d**

1. standard condition, 5h:   n.r.
2. standard condition, 12h:   n.d.

**e**

1. standard condition:                            3 h, 95%
2. standard condition without Na$_2$CO$_3$:   12 h, 28%
3. standard condition without PhI:            12 h, 54%
4. standard condition without [Pd]:           12 h, trace
5. under O$_2$ atmosphere without [Pd]:   12 h, trace

**f**

**Fig. 7 Control experiments. a** The necessity of the hydroxyl group. **b** Transformation of **1-Cs**. **c** Capturing of possible intermediate using nBuCl.
**d** Transformation of **141** in the standard conditions. **e** Transformation of **142** in the comparative conditions. **f** Transformation of **143** in the standard conditions.

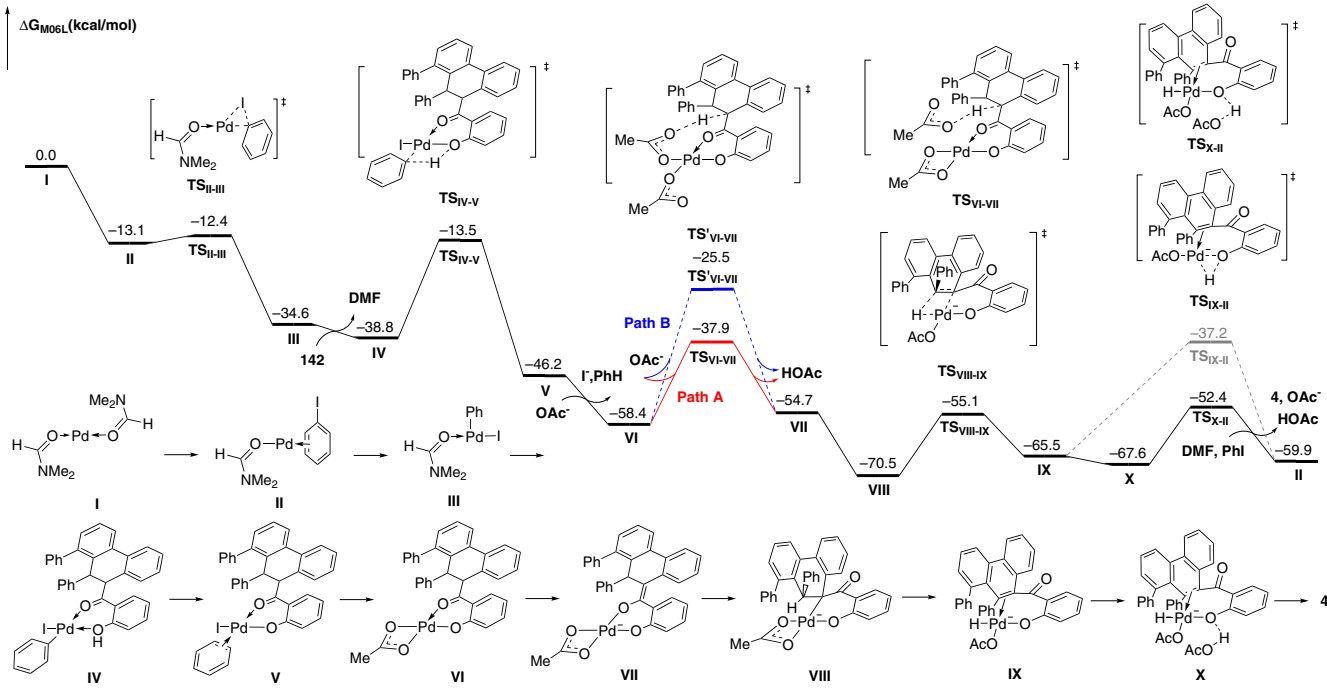

**Fig. 8 Theoretical calculations.** Energy profile for dehydrogenation process of the quadruple C–H activation cascade.

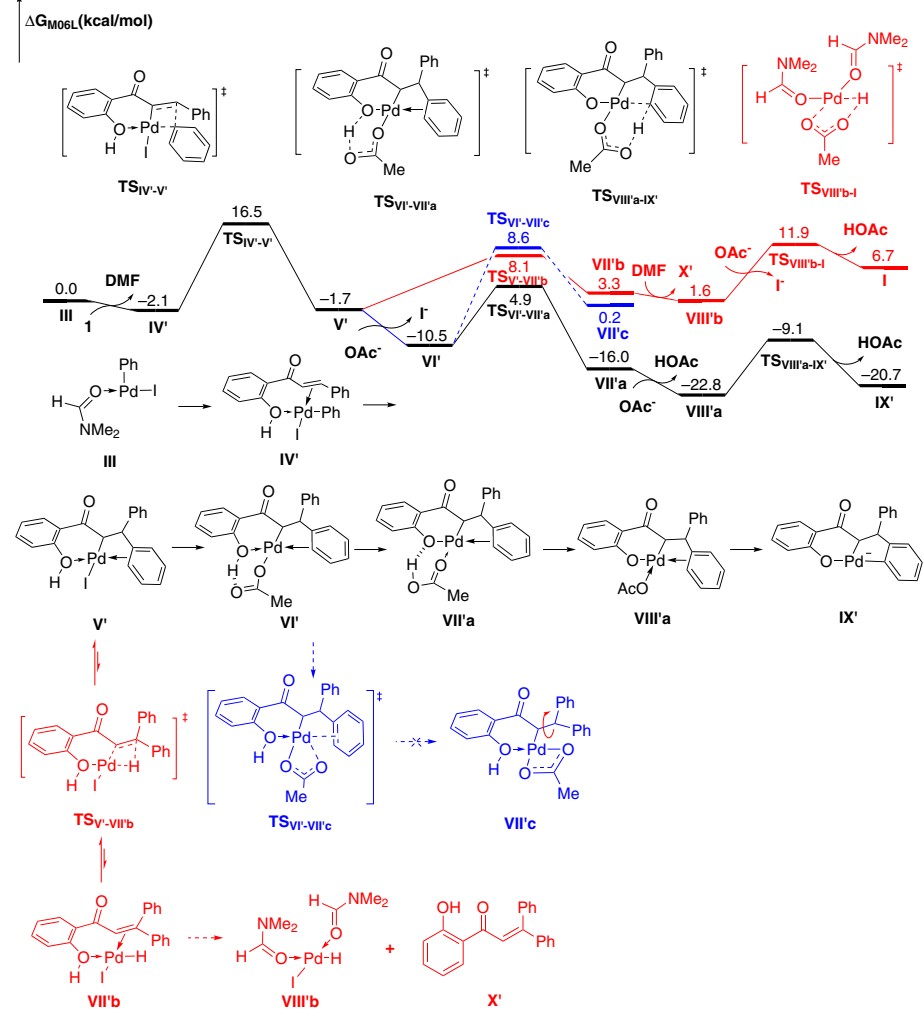

**Fig. 9 Theoretical calculations.** Energy profile for the formation of key six-membered oxo-palladacycle intermediate.

17.6 kcal mol$^{-1}$ more stable than **VIII′b**. Therefore, the β–H elimination pathway is unfavorable from both thermodynamic and kinetic viewpoints. Moreover, when complex **V′** is formed, one phenyl group coming from iodobenzene coordinates onto Pd leading to the discrimination of another one. After a ligand exchange with acetate, a more stable complex **VI′** can be formed, where the coordination of phenyl group is remained. We found that the dissociation of phenyl group by ligand exchange with another oxygen atom in coordinated acetate need to bear a free energy barrier of 19.1 kcal mol$^{-1}$ via transition state **TS$_{VI′–VII′c}$**. As a contrast, the activation free energy for the alternative proton abstraction via a six-membered transition state (**TS$_{VI′–VII′a}$**) to afford complex **VII′a** is only 15.4 kcal mol$^{-1}$. Therefore, the exchange of two phenyl groups cannot be achieved. Next, the extrusion of HOAc affords the highly reactive complex **VIII′a**, followed by a CMD process (**TS$_{VIII′a–IX′}$**) to deliver the key intermediate **IX′**. Note that this C–H activation step is highly feasible, with the energy barrier was calculated to be 13.7 kcal mol$^{-1}$ (for computational details, see Supplementary Data 1).

## Discussion

In conclusion, we have developed a weaker coordination dominated Pd(0)-catalyzed quadruple C–H activation cascade as well as hydro-functionalization, C–H silylation, and C–H borylation sequences with broad substrate scope, good yields, and excellent chemo-selectivity. Relatively weak coordination between the hydroxyl group and metal is the key factor for these cascade transformations. DFT calculations indicate that the β–H elimination is unfavorable based on thermodynamic and kinetic considerations and also rationalize the final dehydrogenation process of the quadruple C–H activation cascade. This strategy represents a generally applicable approach that is compatible with other functionalization processes such as C–H silylation and borylation, which represent a strategic expansion of the Heck reaction. Further studies toward additional potentially useful properties of these compounds are in progress.

## Methods

**General procedure for synthesis of 4.** A 25 ml Schlenk-type tube was charged with the mixture of alkene (0.1 mmol), iodobenzene (0.4 mmol), Pd(OAc)$_2$ (5 mol%), Na$_2$CO$_3$ (0.2 mmol), H$_2$O (40 μL) in DMF (2 mL). The reaction was frozen with the liquid nitrogen and then the tube was evacuated and backfilled with nitrogen for five times. The mixture was first stirred at room temperature for 10 min and then stirred at 110 °C for 12 h. After cooling to room temperature, the mixture was quenched with water, extracted with EtOAc, and concentrated under reduced pressure. The crude product was purified by column chromatography on silica gel (eluent: petroleum ether/EtOAc = 100/1) to afford the products **4**.

## Data availability

The authors declare that all the data supporting the findings of this study are available within the article and Supplementary Information files, and are also available from the corresponding author upon reasonable request. The X-ray crystallographic coordinate for structure reported in this study have been deposited at the Cambridge Crystallographic Data Centre (CCDC) under deposition numbers 1970136 (**3**), 1964456 (**4**), 1961529 (**44**), 1961532 (**50**), 1969923 (**73**), 1969916 (**82**), 1969930 (**103**), 1969931 (**105**), 1974984 (**131**) and 1961844 (**141**). These data could be obtained free of charge from The Cambridge Crystallographic Data Center via http://www.ccdc.cam.ac.uk/data_request/cif.

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

## Acknowledgements

This work was supported by the National Natural Science Foundation of China (Grants 21971080, 21971079, 21602070 and 21772051). This work was supported by 'The Fundamental Research Funds for the Central Universities' (CCNU15ZX002 and CCNU18QN011). This work was also supported by Henan Province Supercomputing Center.

## Author contributions

B.-C.T. discovered the reactions and conducted the experiments; W.-X.L. carried out the DFT calculation. Y.L. and A.-X.W. directed the projects. X.-L.C, C.H, J.-T.M., and Y.-D.W. contributed to the discussion.

## Competing interests

The authors declare no competing interests.
