## [Peer Review File · Nature Communications]

REVIEWER COMMENTS

Reviewer #1 (Remarks to the Author):

In this manuscript, the authors described a novel strategic expansion for the classic Heck reaction. The substrate scope of this reaction was wide, and the products with -carbazole, -TMS and -Bpin are of great interest to the potential applications. Significantly, this reaction provided a new strategy to avoid β -H elimination in Heck reaction. The key six-membered oxo-palladacycle intermediate in this cascade C-H activation reaction was quite interesting and showed different reactivity compared with the well-reported pyridine and aminoquinoline directed reactions. The mechanistic studies and DFT study were impressive and aligned with the proposed pathway. This manuscript is well prepared and is suitable to the readership of Nature Communication. Therefore, I would like to recommend it for publication in Nature Communication after minor revision.

1. In the final product of C-H silylation process, where does the hydrogen at the α -position of carbonyl group come from?
2. In scheme 9f, what is the product?
3. Can this cascade reaction happen if replace the 2'-OH-acetophenone with 2-acetylpyridine?
4. I wonder whether heterocyclic iodide such as 2-Iodofuran and 4-Iodopyridine could work under the same conditions.
5. In the screening conditions section, it's not convenient to review without the corresponding table. For example, I cannot find the structure of 4a in the manuscript. So I suggest the author to give the screening table in manuscript, rather than in SI.
6. In scheme 9a, "yield of 5a", should be "yield of 3a".

Reviewer #2 (Remarks to the Author):

The manuscript describes a cascade reaction initiated by a Pd-catalyzed migratory insertion, but diverted from the normal Heck type reactivity by a strategically placed coordinating group. The reaction is interesting, but I found it very hard to read. The presentation is not scholarly. For example, on page 3, the authors discuss the formation of the side product 4a, but the structure is not shown, you have to dive deep into the SI to find out which structure is meant. There are some easy principles to follow here:

- All structures that are discussed should be depicted, and numbered consecutively in the order they are mentioned in the text (there are large gaps in the numbering, there are no structures 5-8 in the main manuscript).
- There is no reason to use the label "3a" if there is no "3b" in the manuscript. Lettering is good for indicate structures that are variations on a central core, such a labelling scheme could well be used in the manuscript to simplify the drawings, but it has not been done here.

These are things that should have been seen in the editorial office; a manuscript should not be sent to referees unless it fulfills basic scholarly presentation requirements. But I will go on and try to address the content anyway.

The reactions themselves are interesting. The first, the multiple C-H activation, builds complexity in a single reaction, but the scope is of course limited, all three aryls that are added have to be the same. The other two, the tandem silylation or borylation, are synthetically useful in that a handle for further functionalization is introduced.

The authors have done as much mechanistic work as is reasonable at this point, and have come up with hypotheses that are mostly plausible. I can think of several possible alternative mechanisms, so I wouldn't say they have proof, but I wouldn't demand that at this stage. I do have some points I'd like the authors to comment on.

The first aryl insertion creates an apparently symmetrical intermediate, VI' in Scheme 11, where the two aryls seem equivalent. Yet in all the examples, the authors have never shown that the aryl that is present in the original substrate can undergo functionalization. In the cases where the two aryls are not identical, it seems to be always the newly added aryl that is functionalized. Is this really true, or can functionalization of the other aryl be detected? And why is this? If this is really true, then it is not possible that VI' is an intermediate. The aryl must always retain the coordination to Pd if it is selectively functionalized. There is some precedent for this in work of the Heck reaction from Cacchi about a decade ago, but it is not possible just based on Scheme 11.

The silylation and borylation only undergo mono-functionalization, whereas the aryl consistently undergo polyfunctionalization. What is the reason for this drastic difference?

Scheme 10 indicates a proto-dearylation giving benzene as a byproduct. Scheme 12 indicates that the benzene has been detected by GC-MS. This would be unusual, and if true, should be mentioned in the text. In particular, does the amount correspond to the amount of dehydrogenated product? If not, I can think of several alternatives here.

Minor quibbles:

On page 3, the authors state that they see no Heck-type products, but 4a is clearly the result of a Heck reaction, followed by a cyclization, which may have been Pd-catalyzed.

Structure 48a is a product in Scheme 2, but the structure of it is shown in Scheme 3, confusing.

The reaction in Scheme 4 is a hydroarylation followed by a silylation. What is the source of the added hydrogen? Is this the one that was originally on the C-H activated position of the aryl?

In summary, the synthetic effort is impressive and the results are useful, but I'd like to see a more scholarly presentation before giving a final judgement on this.

Reviewer's comments:

Reviewer #1 (Remarks to the Author):

Comments:

In this manuscript, the authors described a novel strategic expansion for the classic Heck reaction. The substrate scope of this reaction was wide, and the products with -carbazole, -TMS and -Bpin are of great interest to the potential applications. Significantly, this reaction provided a new strategy to avoid β -H elimination in Heck reaction. The key six-membered oxo-palladacycle intermediate in this cascade C-H activation reaction was quite interesting and showed different reactivity compared with the well-reported pyridine and aminoquinoline directed reactions. The mechanistic studies and DFT study were impressive and aligned with the proposed pathway. This manuscript is well prepared and is suitable to the readership of Nature Communication. Therefore, I would like to recommend it for publication in Nature Communication after minor revision.

Response: Thanks for your positive comments.

Specific Comments

1. In the final product of C-H silylation process, where does the hydrogen at the α -position of carbonyl group come from?

Response: Thanks for your comments. Based on our isotopic labeling experiments, we found that this added hydrogen probably came from water, rather than DMF or ArI. Besides, the 80% deuteration of D² indicated a reversible β -H elimination might be involved in the catalytic cycle. The following is the corresponding isotopic labeling experiments. The results and data have been added in the revised SI.

Scheme R1.

2. In scheme 9f, what is the product?

Response: Thanks for your comments. Products in scheme 9f probably are the result of self-coupling of PhI, and intramolecular nucleophilic cyclization of 1. Besides, this transformation also afford inseparable complex mixture.

Scheme R2.

3. Can this cascade reaction happened if replace the 2'-OH-acetophenone with 2-acetylpyridine?

Response: Thanks for your advice. We synthesized the corresponding substrate and tested the transformations under the standard conditions. However, no desired products were detected.

Scheme R3.

4. I wonder whether heterocyclic iodide such as 2-Iodofuran and 4-Iodopyridine could work under the same conditions.

Response: Thanks for your advice. We choose 2-Iodothiophene, 3-Iodothiophene, 2-Iodopyridine and 4-Iodopyridine as the representative heterocyclic iodide to tested the reaction since they are readily available. Unfortunately, it seems that this reaction system is not compatible with heterocyclic iodide substrates, with all of these transformations failed. (multi-arylation with 4-Iodopyridine or 3-Iodothiophene gave massive starting materials. Silylation or borylation with 2-Iodopyridine, 4-Iodopyridine gave massive starting materials, whereas massive inseparable complex mixture was observed with 2-Iodothiophene.)

5. In the screening conditions section, it's not convenient to review without the corresponding table. For example, I cannot find the structure of 4a in the manuscript. So I suggest the author to give the screening table in manuscript, rather than in SI.

Response: We are really sorry for our inappropriate presentation of this section. After careful consideration, we move the discussion of reaction condition screening from main article to SI, to simplify the main article.

6. In scheme 9a, "yield of 5a", should be "yield of 3a".

Response: Sorry for this mistake, we have corrected this error in manuscript.

Reviewer #2 (Remarks to the Author):

The manuscript describes a cascade reaction initiated by a Pd-catalyzed migratory insertion, but diverted from the normal Heck type reactivity by a strategically placed coordinating group. The reaction is interesting, but I found it very hard to read. The presentation is not scholarly. For example, on page 3, the authors discuss the formation of the side product 4a, but the structure is not shown, you have to dive deep into the SI to find out which structure is meant.

Response: Thanks for your comments. We are really sorry for our inappropriate presentation of the manuscript. Now we have tried our best to revise the presentation in a more scholarly manner. After careful consideration, we move the discussion of reaction condition screening from main article to SI, to simplify the main article. Besides, we invited Prof. Lyle Isaacs (University of Maryland) to polish our language in the paper, to further improve the quality of manuscript.

There are some easy principles to follow here:

- All structures that are discussed should be depicted, and numbered consecutively in the order they are mentioned in the text (there are large gaps in the numbering, there are no structures 5-8 in the main manuscript).
- There is no reason to use the label “3a” if there is no “3b” in the manuscript. Lettering is good for indicate structures that are variations on a central core, such a labelling scheme could well be used in the manuscript to simplify the drawings, but it has not been done here.

These are things that should have been seen in the editorial office; a manuscript should not be sent to referees unless it fulfills basic scholarly presentation requirements. But I will go on and try to address the content anyway.

Response: Thank you very much for your earnest and careful advice! We apologize for our scholarly presentation errors. After careful consideration, we hope to use Arabic numerals to label all of the structures, because some variations on a central core are more than 26. Now the numbering of structures have been revised.

The reactions themselves are interesting. The first, the multiple C-H activation, builds complexity in a single reaction, but the scope is of course limited, all three aryls that are added have to be the same. The other two, the tandem silylation or borylation, are synthetically useful in that a handle for further functionalization is introduced.

Response: Thanks for your positive comments.

The authors have done as much mechanistic work as is reasonable at this point, and have come up with hypotheses that are mostly plausible. I can think of several possible alternative mechanisms, so I wouldn't say they have proof, but I wouldn't demand that at this stage. I do have some points I'd like the authors to comment on.

Response: Thanks for your comments. We are graceful for the reveiwer's suggestion of making our manuscript more acceptable.

The first aryl insertion creates an apparently symmetrical intermediate, VI' in Scheme 11, where the two aryls seem equivalent. Yet in all the examples, the authors have never shown that the aryl that is present in the original substrate can undergo functionalization. In the cases where the two aryls are not identical, it seems to be always the newly added aryl that is functionalized. Is this really true, or can functionalization of the other aryl be detected? And why is this? If this is really true, then it is not possible that VI' is an intermediate. The aryl must always retain the coordination to Pd if it is selectively functionalized. There is some precedent for this in work of the Heck reaction from Cacchi about a decade ago, but it is not possible just based on Scheme 11.

Response: Thanks for your comments! Actually, this notable selectivity is iure. Almost all of the transformations in Scheme 2-6 afforded single compound with high yields, and the corresponding experiments of column chromatography and NMR spectrum were running smoothly, no any suspected isomer was observed, as per TLC. Besides, the exact structure of some compounds were verified by X-ray (44, 50, 82, 105). Therefore, based on these experimental facts, this reaction has its inherent selectivity for the two aryls.

But theoretically, functionalization on the original aryl is quite possible. Thus we used GC-MS to detect the possible isomer. We take the silylation as the example because of the relatively low boil point of the corresponding compounds. As shown in Scheme R4, when we increased the reaction temperature to 150 °C, we detected a possible trace-amount of isomer 82' (note: this compound can not be seen on TLC, thus we could not separate and verify its exact structure), the ratio was 82 : 82'= 97 : 3. However, in the standard conditions (70 °C, Scheme R5), the ratio was 82 : 82'= 99.1 : 0.9 (the exact structure of 82 was verified by X-ray). This experiment indicates that higher temperature may promote the C-H functionalization on original aryl. However, this process is extremely unfavorable compared with the formation of main product.

Scheme R4.

Scheme R5.

To further explain this selectivity, two competing pathways have been investigated by DFT calculations (Scheme R6). We found that when complex **V'** is formed, one phenyl group coming from iodobenzene coordinates onto Pd leading to the discrimination of another one. After a ligand exchange with acetate, a more stable complex **VI'** can be formed, where the coordination of phenyl group is remained. We found that the dissociation of phenyl group by ligand exchange with another oxygen atom in coordinated acetate need to bear a free energy barrier of 19.1 kcal mol⁻¹ via transition state TS_{VI'-VII'c} (Path A). As a contrast, the activation free energy for the alternative proton abstraction via a six-membered transition state (TS_{VI'-VII'a}) to afford complex **VII'a** is only 15.4 kcal mol⁻¹ (path B). Therefore, the exchange of two phenyl groups cannot be achieved. We have revised this part in new version of draft and highlighted it.

Finally, according to your advice, the structure **VI'** and the corresponding energy in the main article, as well as the corresponding data of DFT calculation in SI, have been revised. Furthermore, we added some discussions about this notable selectivity in the main article (in the discussion of Scheme 11).

Scheme R6.

The silylation and borylation only undergo mono-functionalization, whereas the aryl consistently undergo polyfunctionalization. What is the reason for this drastic difference?

Response: Thanks for your comments. Now we have different understanding of the silylation and borylation process by following your valuable advice (The reaction in Scheme 4 is a hydroarylation followed by a silylation. What is the source of the added hydrogen? Is this the one that was originally on the C-H activated position of the aryl?).

Scheme R7.

Take the silylation process as an example, after our isotopic labeling experiments (Scheme R7), we found that the added hydrogen was probably come from water, rather than DMF or ArI. Besides, the 80% deuteration of D² indicate a reversible β-H elimination might be involved in the catalytic cycle. Based on these findings, we came

up with a new mechanism of the silylation process (Scheme R8). If the silylation process undergoes a di-functionalization (path B), then a TMS anion species would be formed, which is unstable and make this process unfavorable (Scheme R9). Therefore, we considered that the non-reacted part of TMS should be combined with Pd(II) until the reductive elimination of Pd(II), to afford TMSX (X=anion) and Pd(0). Based on this understanding, we proposed that intermediate G undergoes a consecutive protonation to give TMS-Pd(II)-X and final mono-silylation product, and this pathway would be a kinetically favored process.

Scheme R8. Possible Mechanism of Silylation

Scheme R9.

On the other hand, as for poly-functionalization of aryl, after the first o-position functionalization completed, the β -H elimination or protodepalladation process is still unfavorable for both thermodynamic and kinetic aspects (Scheme R10). Comparatively, due to the good leaving ability of iodine anion, the corresponding intermediate would undergoes the second, a very fast another o-position functionalization, which is a kinetically favorable step. Therefore, we considered that the driving force of this domino-type process is very strong.

The added isotopic labeling experiments, corresponding ^1H NMR data and new mechanism of silylation (Page S13-S15 in SI) have been added to the revised SI.

Scheme R10.

Scheme 10 indicates a proto-dearylation giving benzene as a byproduct. Scheme 12 indicates that the benzene has been detected by GC-MS. This would be unusual, and if true, should be mentioned in the text. In particular, does the amount correspond to the amount of dehydrogenated product? If not, I can think of several alternatives here.

Response: Thank you for your comments. There are indeed some gaps of the exact mechanism in dehydrogenation process. Because (1) we could still observed the dehydrogenation product in middle yield without the addition of PhI; (2) we could not detect the corresponding amount of benzene by GC-MS. Despite this, according to the fact that no hydrogen gas was detected during the reaction (SI, Page 7), then the possibility of a oxidative addition process from -O-H to Pd(0) could be excluded. So we considered that the dehydrogenation process was initiated by Pd(II) rather than Pd(0). Therefore, the key issue is to find the real oxidant that oxidize Pd(0) to Pd(II).

In this regard, we think it is reasonable to take the excess PhI as the potential oxidant. The reasons are: (1) In scheme 9e, the yield of dehydrogenation product obviously reduced when we removed the PhI. (2) we do have detected a certain amount of benzene that possibly is a reduction product of PhI.

Actually, PhI is not likely the only oxidant in this process. An exact and complete mechanistic process maybe more complicated than that we considered. But we believe that it is reasonable to consider the excess PhI as a potential oxidant. Finally, according to your advice, we have added some discussions in the main article (reference [22]) about the detection of benzene by GC-MS, as well as the existence of possibility for other plausible oxidative pathways.

Minor quibbles:

On page 3, the authors state that they see no Heck-type products, but 4a is clearly the result of a Heck reaction, followed by a cyclization, which may have been Pd-catalyzed.

Response: Thank you for pointing out our mistake! Now the expression “Heck-type by-products” has been changed to “simple β -H elimination product” (In SI – Table S1 now).

Structure 48a is a product in Scheme 2, but the structure of it is shown in Scheme 3, confusing.

Response: Thank you for pointing out our mistake! Now the crystal structure of 48a (44 now) has been moved to Scheme 2.

The reaction in Scheme 4 is a hydroarylation followed by a silylation. What is the source of the added hydrogen? Is this the one that was originally on the C-H activated position of the aryl?

Response: Thank you for your valuable advice! As we discussed above, the added hydrogen was probably come from water, rather than DMF or aryl.

In summary, the synthetic effort is impressive and the results are useful, but I'd like to see a more scholarly presentation before giving a final judgement on this.

Response: Thank you very much for your constructive advice! Now we have revised the manuscript to make our presentation in a more scholarly manner. We are grateful for your suggestion of making our manuscript more acceptable.

REVIEWER COMMENTS

Reviewer #1 (Remarks to the Author):

The authors have made an improvement for both science and scholarly presentation in the revised manuscript. The questions raised by referees were well resolved and properly represented in either revised manuscript or supporting information. This referee supports the acceptance of this manuscript in its current version.

Reviewer #2 (Remarks to the Author):

The manuscript has been substantially improved, and my requests addressed. The wide scope and the huge amount of experimental work underlying this study is now much more apparent, and the mechanistic arguments more compelling. I am in favor of publication, but have a few minor arguments that I would like the authors to consider.

Page 4, in the range of protected hydroxy groups, the N2BF4 group sticks out. It is not a protected phenol, but might conceivably react with water to produce a hydroxy group in situ. Is this what the authors are proposing?

Page 5 and Scheme 4, when describing the initial reaction leading to 69, there is no explanation at this point where the hydrogen is coming from. By looking at the mechanism, I would assume that the initial product is a silyl enol ether (formed using the second TMS group in the reagent), which is then hydrolyzed to the final product during aqueous workup. The alternative is that there is an undisclosed proton source (maybe a small amount of water) present in the reaction mixture. Could the authors comment on this?

Page 10, the "final dehydrogenation" is in principle a Saegusa-Ito oxidation, with iodobenzene as a slightly unusual stoichiometric oxidant. The reaction has an established mechanism, very much in line with the computational study by the authors. I believe a mention and a reference would be in order. The authors have located one path to the products, which may be sufficient, but my personal opinion is that in a complex reaction mixture like this, there are many other possibilities, I'm not convinced the exact path is what the authors have proposed. One specific problem is that there is an unrecognized Curtin-Hammett situation in VIII-IX-X. Compared to TS(IX-X), the equilibration between VIII and IX is fast, so the barrier must be computed from the lower of them. This is a fundamental misunderstanding which must be corrected; the barrier must always be computed from the lowest preceding point. Thus, the barrier is really 33.3 kcal/mol, which is too high to be believable. I am sure there is a lower energy path from VIII to the final products, possibly involving reprotonation using the acetic acid formed together with VII.

Per-Ola Norrby

Response to Manuscript ID: NCOMMS-20-19268-A

Reviewer #2 (Remarks to the Author):

The manuscript has been substantially improved, and my requests addressed. The wide scope and the huge amount of experimental work underlying this study is now much more apparent, and the mechanistic arguments more compelling. I am in favor of publication, but have a few minor arguments that I would like the authors to consider.

Response: Thanks again for your positive comments for our work, and we will try our best to refine the manuscript by following your professional advice.

Page 4, in the range of protected hydroxy groups, the N₂BF₄ group sticks out. It is not a protected phenol, but might conceivably react with water to produce a hydroxy group in situ. Is this what the authors are proposing?

Response: Thanks for your careful advice! Now we have removed this non-rigorous expression in the main article.

Page 5 and Scheme 4, when describing the initial reaction leading to 69, there is no explanation at this point where the hydrogen is coming from. By looking at the mechanism, I would assume that the initial product is a silyl enol ether (formed using the second TMS group in the reagent), which is then hydrolyzed to the final product during aqueous workup. The alternative is that there is an undisclosed proton source (maybe a small amount of water) present in the reaction mixture. Could the authors comment on this?

Response: Thanks for your comments. Actually, based on our understanding of this reaction, we considered that there were several possible pathways that existed simultaneously of the completed process from **H** to **69** (Scheme R1). Therefore, according to your advice, we selected two representative possible pathways to investigate (Scheme R2). One is hydrolysis via 1,3-palladium shift, and silyl enol ether intermediate (Path A, Scheme R2-a). The other is direct protodepalladation to afford **69** and Pd(II) species (Path B, Scheme R2-b).

We performed the reaction in *dry*-DMF with H₂O¹⁸ (6eq), and the possible O¹⁸-labeled product (approx. 24%) was detected, as per MS (Scheme R3). This result indicated that both pathway A and B may exist, and in particular, we could not detect the silyl enol ether intermediate (Scheme-R2-a, Int. **H-b**) by lowering the temperature or shortening time, as per TLC and GC-MS, which suggested that the hydrolysis of Int. **H-b**, may have happened during the reaction. Therefore, at this stage, we speculate that the direct protonation (Path B) to afford the final product is more favorable. Besides, water (in the reaction system) is most likely the origin of the added hydrogen for the protonation, based on our previous mechanistic research (Scheme R4).

Finally, according to your advice, we added some comments in the main article, about the origination of the added hydrogen in silylation.

Scheme R1

Path A: Hydrolysis via 1,3-palladium shift, and silyl enol ether intermediate.

Scheme R2-a

Path B: Direct protodepalladation.

Scheme R2-b

Scheme R3

Scheme R4

Page 10, the “final dehydrogenation” is in principle a Saegusa-Ito oxidation, with iodobenzene as a slightly unusual stoichiometric oxidant. The reaction has an established mechanism, very much in line with the computational study by the authors. I believe a mention and a reference would be in order. The authors have located one path to the products, which may be sufficient, but my personal opinion is that in a complex reaction mixture like this, there are many other possibilities, I’m not convinced the exact path is what the authors have proposed. One specific problem is that there is an unrecognized Curtin-Hammett situation in VIII-IX-X. Compared to TS(IX-X), the equilibration between VIII and IX is fast, so the barrier must be computed from the lower of them. This is a fundamental misunderstanding which must be corrected; the barrier must always be computed from the lowest preceding point. Thus, the barrier is really 33.3 kcal/mol, which is too high to be believable. I am sure there is a lower energy path from VIII to the final products, possibly involving reprotonation using the acetic acid formed together with VII.

Response: Thanks so much for this comment. We also mentioned this unusual activation free energy for this step. According to the useful comment, we have done some additional calculation. We found that the protonation of Pd-phenolate complex could be realized by using an extra acetic acid. DFT calculation found that the activation barrier is reduced to 15.2 kcal mol⁻¹ following this idea. Therefore, we revised the first paragraph of Page 11 as: “Then a syn-β-H elimination affords a Pd(II)-hydride complex IX. Interestingly, the direct O-H reductive elimination need to bear a free energy barrier of 28.3 kcal mol⁻¹. Alternatively, anionic complex IX can form a hydrogen bond with acetic acid to form complex X reversibly. Sequentially, a proton transfer takes place via transition state TS_{X-II}. The calculated free energy barrier for this step is only 15.2 kcal mol⁻¹. After ligand exchange, product 4 could be yielded with the regeneration of catalytically active complex II.” (Scheme R5).

Scheme R5

Finally, according to your advice, we added some comments and corresponding references about the Saegusa-Ito oxidation in the part of “final dehydrogenation”.

REVIEWERS' COMMENTS

Reviewer #2 (Remarks to the Author):

The authors have addressed all my comments. I believe the manuscript is now suitable for publication.

/Per-Ola Norrby

Response to Manuscript ID: NCOMMS-20-19268-B

Reviewer #2 (Remarks to the Author):

The authors have addressed all my comments. I believe the manuscript is now suitable for publication.

Response: Thanks again for your professional advice to our work.